# Endometriosis Stem Cells as a Possible Main Target for Carcinogenesis of Endometriosis-Associated Ovarian Cancer (EAOC)

**DOI:** 10.3390/cancers15010111

**Published:** 2022-12-24

**Authors:** Jacek R. Wilczyński, Maria Szubert, Edyta Paradowska, Miłosz Wilczyński

**Affiliations:** 1Department of Surgical Gynecology and Gynecological Oncology, Medical University of Lodz, 90-419 Lodz, Poland; 2Laboratory of Virology, Institute of Medical Biology, Polish Academy of Sciences, 93-232 Lodz, Poland; 3Department of Surgical Gynecology, Endoscopic Gynecology and Gynecological Oncology, Polish Mother’s Health Center—Research Institute, 93-348 Lodz, Poland

**Keywords:** endometriosis, endometriosis stem cells, stem cells, endometriosis-associated ovarian cancer, ovarian cancer

## Abstract

**Simple Summary:**

Endometriosis is a risk factor for some histologic types of ovarian cancer defined as endometriosis-associated ovarian cancers (EAOC). Although the mechanism promoting the carcinogenesis leading from endometriosis into the EAOC has been not completely known, the possible role of endometriosis stem cells (ESCs) in this process may be important and is discussed in this article.

**Abstract:**

Endometriosis is a serious recurrent disease impairing the quality of life and fertility, and being a risk for some histologic types of ovarian cancer defined as endometriosis-associated ovarian cancers (EAOC). The presence of stem cells in the endometriotic foci could account for the proliferative, migrative and angiogenic activity of the lesions. Their phenotype and sources have been described. The similarly disturbed expression of several genes, miRNAs, galectins and chaperones has been observed both in endometriotic lesions and in ovarian or endometrial cancer. The importance of stem cells for nascence and sustain of malignant tumors is commonly appreciated. Although the proposed mechanisms promoting carcinogenesis leading from endometriosis into the EAOC are not completely known, they have been discussed in several articles. However, the role of endometriosis stem cells (ESCs) has not been discussed in this context. Here, we postulate that ESCs may be a main target for the carcinogenesis of EAOC and present the possible sequence of events resulting finally in the development of EAOC.

## 1. Introduction

Endometriosis is an inflammatory, estrogen-dependent disease characterized by the presence of both endometrial epithelium and stroma outside the uterine cavity [1]. The term “endometriosis implant”, “endometriosis focus” or “endometriosis lesion” is histologically identical and exchangeable with the term “ectopic endometrium” and is used in this context further in the text. The term “eutopic endometrium” characterizes the endometrial tissue localized normally inside the uterine cavity. The real frequency of endometriosis in the population of reproductive-age women is unknown but estimated to be approximately 10%. However, it is much more frequent in a population of infertile women (estimations vary from 20–50%) [2]. According to the localization endometriosis could be classified as peritoneal, ovarian (endometriotic or chocolate cysts), deep-infiltrating endometriosis DIE (inter-organ spaces or organ walls usually in the pelvis), endometriosis localized in distant places (i.e., lungs, brain, nose) and iatrogenic endometriosis of abdominal wall in the proximity of the scar after cesarean section [3,4]. The typical syndromes of endometriosis are dysmenorrhoea and painful intercourses, however, up to 25% of affected patients are asymptomatic [5]. Endometriosis also impairs fertility, especially when presented in an advanced stage which negatively influences oocyte quality, tube motility and implantation environment [6]. The predisposition to endometriosis seems to be hereditary, as female first-degree relatives of patients with recognized endometriosis have a six times higher risk for the disease [7]. Genome-wide associated studies (GWAS)s identified single-nucleotide polymorphisms (SNPs) associated with endometriosis risk; [8,9,10]. The identified loci-containing genes were responsible for stem cell function (WNT4), ovulation (FSHB, ESR1), and estrogen signaling (ESR1, GREB1, CCDC170, CYP2C19) [3]. These associations were also noticed for both breast and ovarian cancer. However, no germline mutations of these genes have been noted in familial cases of endometriosis. It seems that the pattern of inheritance is not dependent on a single gene. Instead, it seems that the etiology is multifactorial and genetic, epigenetic and environmental factors all contribute to the occurrence and progression of the disease [11]. 

An association between endometriosis and increased risk for ovarian cancer has been demonstrated by a large registry study of Swedish women. The standardized incidence ratio SIR was 1.9 for women with a follow-up exceeding 10 years, and the risk was higher in patients with early diagnosed or long-lasting endometriosis (SIR 2.01 and 2.23, respectively) [12,13]. Another large epidemiological study found an increased risk for ovarian cancer with an odds ratio OR 1.46, and a significant association between endometriosis and defined histological sub-types of ovarian cancer, mainly clear-cell ovarian cancer (CCOC) OR 3.05, endometroid ovarian cancer (ENOC) OR 2.04 and finally low-grade serous ovarian cancer (LGSOC) OR 2.11. The history of endometriosis is associated with a lifetime risk of 1.5% of developing these types of cancer [14]. However, endometriosis is not a risk factor for high-grade serous ovarian cancer (HGSOC) and mucinous ovarian cancer, which suggests a different etiology of these tumors. In the group of patients operated on because of CCOC or ENOC, the incidence of concomitant endometriosis was 26% and 21%, respectively, while for other ovarian cancer types incidence did not exceed 6% [15]. The atypical endometriosis in ovarian cysts has been suggested as a precursor lesion for both CCOC and ENOC [16,17]. Another possibility is the malignant transformation of endometriotic lesions localized on the ovarian surface or the malignant transformation of tubal endometriosis [18]. The coexistence of tubal endometriosis and endosalpingiosis (the presence of ectopic tubal epithelium) was also found to be a risk factor for EAOC [19]. 

The etiology of endometriosis has not been completely elucidated. The most accepted theory has been proposed by Sampson, who suggested that retrograde menstruation transported exfoliated endometrial cells into the peritoneal cavity where they were able to implant and grow to form endometrial lesions [20]. Sampson’s theory explains peritoneal and ovarian endometriosis, however, does not explain the presence of DIE or endometriosis in remote localizations, as well as cannot explain endometriosis after the hysterectomy or in the rare cases of endometriosis in men subjected to the hormonal treatment of prostate or bladder cancer [21,22,23]. Lymphatic or hematogenous spread of endometriosis cells could explain DIE or remote localizations, but in the remaining cases, metaplasia from progenitor stem cells could be a possible explanation. 

The presence of stem cells inside the endometrium and endometriotic lesions, and the connection between endometriosis and EAOC cancers pose the question about the possible involvement of endometriosis stem cells in the carcinogenesis of EAOC tumors. 

## 2. Stem Cells Inside the Endometrium

Adult stem cells are multipotent cells, which means that they are capable to differentiate into several but not all cell types (usually within the same germ lineage) and to renew their own population. The human endometrium regenerates in the menstrual cycle as well as after labor, which indicates the presence of the cell population having stem cell properties. The characterization of this population has not been well defined, however, there are several candidate populations described. Endometrial stem cells could be derived either from adult stem cells residing locally inside the endometrium or from bone marrow-derived stem cells which home the endometrium in response to physiological (menstruation) or pathological injury [24]. 

### 2.1. Epithelial Stem Cells

The evidence of putative endometrial and epithelial stem cells was first shown by Chan et al. in 2004 [25]. They described small populations of cells present inside endometrial epithelium and stroma and comprising 0.22% and 1.25% of cells, respectively. These populations showed clonogenicity and regenerative properties. Transforming-growth factor-α (TGF-α), epidermal growth factor (EGF), platelet-derived growth factor (PDGF), leukemia-inhibitory factor (LIF), hepatocyte growth factor (HGF), stem cell factor (SCF) and insulin-like growth factor-1 (IGF-1) supported clonogenicity of epithelial cells, while TGF-α, EGF, PDGF and basic fibroblast growth factor (bFGF) supported clonogenic stromal cells [25]. Clonogenic stromal cell cultures contained both fibroblast and myofibroblast cells. Gargett et al. described the presence of epithelial EpCAM^+^ progenitor cells showing significant clonogenicity and the ability to reconstitute epithelial glandular pattern [26]. These cells possessed self-renewal capability, high proliferative potential and were rather unipotential. Similar populations of epithelial N-cadherin+, stage-specific embryonic antigen-1 (SSEA-1/CD15)^+^ or Axin2^+^ stem cells/ progenitors capable to form glandular lineages have been described by others [27,28]. Further studies showed that endometrium contained the progenitor cells possessing stem cell properties and contributed to endometrial growth which was characterized by the presence of stemness markers including transcription factor OCT-4, CD117, CD34 and endometrial carcinoma protein Musashi-1 [29,30,31,32,33,34,35,36]. Molecules OCT-4 and CD117 are considered stemness-related markers for many different cells [37]. Marker CD34 is expressed on both endothelial and epithelial progenitors, and some mesenchymal stem cells (MSCs) [30,31,38,39]. 

### 2.2. Perivascular Progenitor/Stem CD34^+^KLF-4^+^ Cells

Another marker of stem endometrial cells could be the transcription factor Kruppel-like factor-4 (KLF-4) found to be up-regulated in some progenitor CD34^+^ cells [38]. The population of CD34^+^KLF-4^+^ progenitor/stem cells located inside endometrial stroma in perivascular regions has been noticed. These cells proliferate vigorously and migrate to the sites of endometrial injury [40]. Small ubiquitin-like modifiers (SUMO) are a group of proteins that through covalent attachment to different proteins (called SUMOylation) are capable to modify their function, mainly in processes of intracellular transport, transcription regulation, apoptosis and cell proliferation. SUMOylation is reversed by SUMO endopeptidases (SENPs) [41,42,43]. SUMOylation has been connected to the regulation of stemness by influencing transcription factors OCT-4, SOX-2 and NANOG and could be engaged in endometrium decidualization [44,45,46]. SUMOylation of estrogen receptor-α (ER-α) up-regulates proliferative activity of CD34^+^KLF-4^+^ progenitor/stem cells and supports endometrial regeneration [40]. SENP1deletion increases ER-α SUMOylation resulting in endometrial hyperplasia or endometrial cancer as shown in mice with a stromal SM22α-specific SENP1 deletion (SENP1 smKO). Moreover, in SENP1smKO mice, delayed oocyte growth and follicle maturation were observed [47]. 

### 2.3. “Side Population” Cells

Cells of stem-like properties have been also isolated from short-time cultured endometrial cells and characterized as classical “side population” (SP) cells. This population is defined as a population of cells displaying low Hoechst-33342 dye fluorescence in cytometric analysis and possessing the functional properties of recovery of the tissue of cell origin [48]. Human endometrium contains approximately 1–7% of SP cells [33]. Studies indicated that SP cells are more heterogenic than initially has been thought, and originate from the endothelial lineage (CD31^+^), hematopoietic lineage (CD34^+^CD45^+^), epithelial lineage (EMA^+^) and mesenchymal lineage (CD90^+^CD105^+^CD146^+^) [49,50]. Endometrial SP cells were characterized as having either epithelial or stromal origin. Epithelium-originated SP cells indicated CD9^+^CD90^+^CD105^+^CD73^+^ CD45^+^CD34^+^CD31^+^CD133^+^STRO-1^+^ phenotype and capability to form adipocytes and osteocytes. Stroma-originated SP cells showed CD90^+^CD73^+^CD45^+^ CD34^+^CD31^+^CD133^+^STRO-1^+^Vimentin^+^ phenotype and also differentiated to adipocytes or osteocytes [50]. Another population of endometrial SP cells was phenotyped as CD9^−^CD13^−^CD45^−^CD34^−^ABCG2^+^ cells and were isolated in the highest numbers during menstruation and from early proliferative phase endometrium [51]. In vitro differentiation of endometrial SP cells resulted in the growth of CD13^+^ stromal endometrial cells, CD31^+^ endothelial cells and cytokeratin+ endometrial epithelial cells [52]. In the xenotransplantation model endometrial SP cells were able to reconstitute endometrial tissues in NOD/SCID/γ(c)null mice including stromal, epithelial and vascular components [53]. Endometrial SP ABCG2^+^ cells were localized not only inside the basalis layer of endometrium but also in its functional layer, and were preferentially situated in small capillaries which could suggest their bone-marrow origin [51]. The distribution of estrogen receptors differs between endometrial finally differentiated cells and endometrial SP cells. While differentiated cells express ER-α receptor, SP cells show mainly the expression of ER-β (similarly to bone marrow-derived endothelial progenitor cells) [54]. The highly ER-β expressive endometrial SP cells could be involved in the pathogenesis of aggressive endometrial adenocarcinomas as well as in endometriosis, as they show remarkable proliferative, migratory and angiogenic activity [55,56]. Aldehyde dehydrogenases (ALDH) are considered markers of both normal tissue and CSCs. ALDH1A1 and ALDH1A3 are expressed in the epithelium of the basalis layer of eutopic endometrium, showing that progenitor/stem cells are present in eutopic endometrium and play a role in its physiology [57]. The identification of cells with high ALDH expression is associated with poor outcomes in several gynecologic malignancies [58]. Because ALDH isoforms contribute to the stemness of cancer cells, therapies using different ALDH inhibitors to target CSCs are needed.

### 2.4. Bone Marrow-Derived Stem Cells (BMDSCs)

Another population of endometrial stem cells is the bone marrow-derived stem cells (BMDSCs) [26]. The presence of this population was confirmed in the endometrium of bone marrow human female recipients and in a mouse model where recipient female mice transplanted with the bone marrow-derived from male donors showed the Y chromosome-positive BMDSCs [59,60]. The BMDSCs constituted around 1–8% of epithelial and 8–10% of stromal cells in the endometrium, respectively [61]. They were recruited toward the sites of endometrial injury and were probably responsible for the regrowth and regenerative properties of the endometrium [62]. MSCs are an example of such BMDSCs multipotent cells which have been identified in the bone marrow and endometrium [26,63,64]. The presence of stromal cells of MSCs phenotype (CD29^+^CD44^+^CD73^+^CD90^+^CD105^+^CD140b^+^CD146^+^CD31-CD34-CD45-) was first observed by Gargett et al. [26]. The cells were multipotent and differentiated into adipocytes, chondrocytes, myocytes and osteocytes. Gurung et al. have described the population of perivascular CD146^+^PDGFRβ^+^, known also as CD146^+^CD140b^+^ cells showing the characteristics of MSCs with the same phenotype and differentiation capability as the stromal cells described by Garrett et al. [65]. Resident perivascular cells like CD34^+^KLF-4^+^ or CD146^+^PDGFRβ^+^ could both be the source of endometrial stem cells [40]. CD146^+^PDGFRβ^+^ cells express strongly genes associated with angiogenesis, hypoxia, steroid hormone response, inflammation, and stemness (NOTCH, Hedgehog, IGF) [66]. Interleukin-6, CXCL1 and CXCL5 increase the proliferation and self-renewal potential of CD146^+^PDGFRβ^+^ cells [67]. A novel marker of endometrial MSCs is the sushi domain containing 2 (SUSD2^+^) molecules. The SUSD2^+^ MSCs are localized in perivascular space in both basal and functional layers of the endometrium. They express MSCs markers, like CD29, CD44, CD73, CD90, CD105, CD117, CD140b, CD146 and STRO-1, while lacking the expression of CD31 and CD45 [68]. They can differentiate into adipocytes, myocytes, chondrocytes, osteocytes and endothelial cells. SUSD2^+^ MSCs cells play important role in the decidualization of the endometrium [69]. While TGF-β inhibits proliferation and colony-forming efficiency of SUSD2^+^ MSCs cells, Sonic hedgehog (SHH) signaling augments the regenerative abilities of these cells [70,71]. SUSD2^+^ MSCs cells could effectively influence immunity. They inhibit the maturation of dendritic cells (DCs), stimulate tolerogenic CD4^+^CD25^+^FoxP3^+^ T-regulatory cells (Tregs), CD19^+^CD10^+^ B-regulatory cells (Bregs) and M2-type macrophages. They also decrease the secretion of pro-inflammatory cytokines, like IL-1β, TNF-α and IL-6 [51]. In the mice model, the treatment of Asherman’s syndrome was performed using the BMDSCs, followed by a successful pregnancy in 90% of treated animals vs. 30% of untreated [72]. An experimental treatment has also already been conducted in women with Asherman’s syndrome, with a very promising pregnancy rate [73]. 

### 2.5. Menstrual Stem Cells (MeSCs)

The most active part of the endometrium in the context of regenerative ability is the basalis layer of the endometrium which stays unshed during the menstrual cycle, however, endometrium-derived stem cells (EDSCs) were found also in the menstrual blood [74]. Even after menopause, a small population of stromal cells with MSC properties can be found [75]. The phenotype of menstrual stem cells—MeSCs is characterized by the presence of MSCs and stem cell markers like CD9, CD29, CD41a, CD44, CD59, CD73, CD90, CD105, SSEA-4 and NOTCH1, while their multipotent functions were confirmed by differentiation into endothelial, pancreatic, hepatic, adipocytic and neurocytic cells [76]. EDSCs are also capable to secrete growth factors including PDGF, EGF, vascular-endothelial growth factor (VEGF) and metalloproteinases, as well as indicate pro-angiogenic potential [77]. EDSCs have a higher proliferative potential than BMDSCs [54]. Similarly to SUSD2^+^ cells, EDSCs have a pro-tolerogenic and inhibitory activity on immune cells [78,79]. 

In summary, the landscape of stem/stem-like cells in the endometrium is quite complex. Data are supporting the evidence of both intrauterine and extrauterine unipotent or multipotent cells. They are localized in the endometrial epithelium and inside the stroma. The epithelial cells have been identified as SP epithelial-originated endometrial cells or epithelial progenitor/stem cells. The first ones behave as multipotent cells having phenotype suggesting extrauterine hematopoietic or vascular-associated origin (CD31^+^CD34^+^CD45^+^). The second ones seem to be unipotent (only epithelial glandular lineages) intrauterine cells. Stroma-derived stem cells have characteristics of either fibroblasts (intrauterine origin) or bone marrow MSCs and are multipotent. They reside inside the stromal compartment of the endometrium, and in the case of bone marrow-derived cells, they collect exclusively around or even inside microvessels.

## 3. Stem Cells and Endometriosis

### 3.1. Endometrium-Derived Stem Cells

Endometrium-derived stem cells could be responsible for the generation of endometriosis being shed during menstruation into the peritoneal cavity, where they could be implanted, or transported via hematogenous/lymphatic spread. It was noticed that women with endometriosis showed more progenitor stem cells originating from the basalis layer of endometrium compared to healthy women [80]. Expression of SSEA-1 in ectopic epithelium corresponds to the expression observed in the basalis layer of eutopic endometrium that confirms retrograde menstruation theory [81]. 

The eutopic endometrium of women with endometriosis in the mid-secretory phase expressed significantly higher levels of CD44^+^ cells including CD44v6 molecules. The concentration of soluble CD44 in the serum and endometrial fluid of endometriosis patients was higher than that of healthy women [82]. In the mouse model of endometriosis, the role of epigenetic regulation of B-cell lymphoma 9 (BCL9)/Wnt/β-catenin/CD44 signaling pathway indicated, that exosomes containing miR-30c were able to inhibit invasion and migration of endometrial epithelial cells in ectopic lesions by blocking BCL9/Wnt/CD44 axis [83]. 

Expression of stemness marker OCT-4 was confirmed in epithelial cells of endometriotic lesions and was found to be increased in endometriosis compared to ectopic endometrium (32% vs. 3.5%, respectively) [84,85]. OCT-4 mRNA expression was also increased in endometriosis compared to eutopic endometrium in patients and healthy controls. Moreover, mRNA expression was correlated with the expression of several migration-associated genes [86]. Another transcription factor associated with cellular stemness, SOX-2, was found to be highly expressed in the stromal component of ectopic endometrium. Increased expression of RNA-binding protein Musashi-1 associated with the proliferation of neural and epithelial progenitor cells was noted inside ectopic endometrial lesions [35]. Other markers of cell stemness, CD117, NANOG, NOTCH and NUMB were also up-regulated in endometriosis implants [84,85]. The cells of stem cell properties were isolated from ovarian endometriomas. They expressed stemness markers (SALL4, CD133 and MSI-1 molecules). Cells originated from both epithelial (EpCAM^+^cytokeratin^+^α6-integrin^+^) and stromal (CD90^+^CD10^+^ fibroblast markers) lineage (0.09% and 0.13% of endometrioma cells, respectively) and showed expression of ER-α and ER-β receptors. Cells showed colony-forming and self-renewal capacity in functional tests. Moreover, endometriotic stromal cells were able to differentiate in vitro into mesenchymal cell lineages, including adipogenic, myogenic, osteogenic and chondrogenic ones [25]. Isolated from ovarian endometrioma MSCs were stimulated by 17β-estradiol to form colonies and proliferate. Moreover, upon stimulation, they showed increased expression of OCT-4, CD133 and ALDH1 stem markers. Expression of the nuclear transport receptor importin-13 which is responsible for stem cell differentiation and transcriptional regulation of cellular response against hypoxia was found to be increased in ectopic endometrium [87]. Recent investigations showed evidence of ALDH1 family molecules in endometriotic lesions. ALDH1A1, ALDH1A2 and ALDH1A3 molecules were expressed in the epithelium of ovarian endometrioma posing the possibility that ALDH1^+^ stem cells could play a role in endometriosis pathophysiology [57]. Immunoreactivity of endometriotic epithelial cells to EpCAM and N-cadherin was significantly higher compared to eutopic endometrium, which suggests that overexpression of EpCAM is engaged in Epithelial-Mesenchymal Transition (EMT) in endometriosis [88]. 

Menstrual stem cells (MeSCs) from women with endometriosis indicate increased proliferative and invasive activity compared to MeSCs of healthy women. MeSCs from endometriosis patients show also up-regulated CD9, CD10, CD29, indoleamine 2,3-dioxygenase-1 (IDO-1), COX-2, IL10, IFN-γ and monocyte chemoattractant protein-1 (MCP-1) expression [89].

### 3.2. Tubal-Derived Stem Cells

Endometriosis of fallopian tubes is a frequent reason for tubal obstruction, adhesions and hydrosalpinx resulting in considerable subfertility. The occurrence of tubal endometriosis is estimated to be 0.3% to 14% [90,91]. Paik et al. first identified epithelial stem-like cells that were concentrated in the distal end of the tubes and expressed CD44, EpCAM and integrin α 6 [92]. These cells were capable of multi-lineage differentiation and self-renewal in vitro. The fimbrial part of the tubes contains a population of cells expressing markers of stemness including leucine-rich repeat-containing G-protein-coupled receptor-5 (LGR-5), which is a known marker of stem cells in various epithelia, as well as CD44, SSEA-4 and ALDH molecules. These cells can recapitulate the epithelium of the fimbrial end of the tube and are subject to hormonal regulation by estrogen and progesterone [93,94]. Inflammatory reaction in the tubes and the pelvis induces the proliferation of the epithelium inside the tubes, stimulates the stemness of epithelial stem cells, and promotes the growth of tubal endometriosis [95]. Moreover, chronic infection of the fallopian tube organoids with Chlamydia trachomatis leads to increased epithelial cell proliferation and stemness [95]. Shedding of tubal endometriosis into the peritoneal cavity and ovarian surface could be a risk factor for EAOC, and the importance of stem cells from tubal endometriosis in this phenomenon could not be excluded [18].

### 3.3. Bone Marrow-Derived Stem Cells (BMDSCs)

Another source of stem cells inside endometriotic lesions could be bone marrow which produces MSCs that circle into peritoneal and remote locations and augment the formation of endometriotic lesions [96]. Human endometrial MSCs were isolated from both eutopic and ectopic endometrium in endometriosis patients, however, MSCs from endometriotic lesions showed increased migration, proliferation, invasiveness and vasculogenic potential in comparison to processes described in normal endometrium—as mentioned in previous chapter [97]. Sites of injury and inflammation in the uterus and in endometriosis lesions attract BMDSCs. The recruitment of BMDSCs stem cells into endometriosis ectopic sites is dependent on estrogen and is a very effective phenomenon, as ectopic lesions can attract BMDSCs more than eutopic endometrium [98,99]. 

BMDSCs secrete cytokines that promote the proliferation of ectopic endometrial lesions, which in turn stimulate BMDSCs differentiation [100,101]. Infiltration by both immune and BMDSC cells is mediated by the pro-inflammatory microenvironment of peritoneal fluid [102]. Vascularization of endometriotic lesions is mediated at least partly by endothelial progenitor cells (EPCs) originating from hematopoietic stem cells and bone marrow progenitors [103]. Mobilization of EPCs occurs under the regulation of VEGF and fibroblast growth factor (FGF) which are up-regulated in endometriotic implants. The process is estradiol-dependent [104]. The presence of SUSD2^+^ MSCs stem cells was confirmed in endometriotic implants at a higher frequency than in eutopic tissue. They are engaged in the net of interactions between inflammatory cytokines from peritoneal fluid (IL-6, IL-8, TNF-α) and indicate increased expression of activin A-specific receptor (ALK4) and connective tissue growth factor (CTGF). Activin A is a member of TGF-β cytokine family responsible for the regulation of inflammation, fibrosis and wound repair. Its expression was revealed in inflammatory and autoimmune diseases, as well as in several malignancies [105]. CTGF factor plays a role in cell proliferation, migration, angiogenesis, wound repair, fibrosis and carcinogenesis [106,107]. There is a strong supposition that MSCs/activin A/ CTGF network could be at least one of the mediators of peritoneal inflammation and secondary fibrosis with adhesions formation. Its role in endometriosis-associated carcinogenesis should also be taken into account. The up-regulated expression of activin A may be involved in carcinogenesis by reducing TGF-β-mediated signals inhibiting cell growth in human endometrial adenocarcinoma tissues [108]. Loss of CTGF function may be a factor in the carcinogenesis of ovarian cancer in early stages of a tumor, while in endometrial cancer high CTGF expression was an independent risk factor for a worse prognosis [109,110]. The molecular pattern of stem cells from ectopic lesions differs compared to the pattern displayed by stem cells from eutopic endometrium. Ectopic stem cells showed reduced expression of PTEN, ARID1A and TNF-α, and abnormal expression of c-kit, HIF-2α and E-cadherin [111]. Down-regulation of PTEN expression was also frequently observed in both endometroid and clear-cell endometriosis-associated ovarian cancers [112]. Loss of ARID1A expression was found in high frequency in endometrial endometroid cancer and in EAOC [113].

### 3.4. Endomeriosis Stem Cells Movement

Stem cells are also able to move between endometrium and endometriosis lesions. In the mice model the presence of cells from endometriotic foci of green fluorescent protein (GFP) transgenic donor mice was confirmed in the ectopic endometrium of recipient mice [114]. These cells showed a distinct genetic profile compared to normal endometrium, showing activation of genes responsible for epithelial-to-mesenchymal transition (EMT). However, stem cells were homing endometrial stroma instead of epithelium, which disrupted the endometrial receptivity through the change in the Wnt-signaling in the cells which could have a potentially negative influence on implantation. In humans, advanced endometriosis is one of the main causes of implantation failure [115].

The traffic of stem cells from bone marrow and between eutopic and ectopic localization depends on the function of chemotactic molecules. Endometrial stromal cells express CXCL12, while BMDSCs express its receptor, CXCR4. The existence of the CXCR4-CXCL12 attracting pathway was confirmed earlier in cancer and places where inflammation or tissue injury occurred. Signaling via CXCR4-CXCL12 pathway influences the expression of metalloproteinases and VEGF [116,117]. The function of the CXCR4-CXCL12 pathway was confirmed in vitro where estradiol-stimulated endometrial stromal cells and BMDSCs showed induced expression of chemokines followed by chemoattraction of BMDSCs towards endometrial stromal cells [55]. The activity of the CXCR4-CXCL12 pathway in endometriosis foci was significantly increased [118], therefore, endometriosis successfully competed with eutopic endometrium for the BMDSCs. The endometriotic cells are also present in the blood of endometriosis patients as circulating endometrial cells (CECs), which show characteristics of stem-like cells [119]. 

In summary, endometriotic lesions contain cells of epithelial and stromal lineage showing markers of stemness, like OCT-4, SOX-2, MSI-1, NANOG, NOTCH, NUMB, SALL-4, CD133 or CD117. There are also bone marrow-derived BMDSCs cells with are attracted in greater numbers to ectopic than to eutopic endometrium and show increased migration, proliferation and angiogenic properties. They also orchestrate with the peritoneal net of inflammatory cytokines and growth factors. The expression of several genes and molecules responsible for the regulation of endometriotic implant growth is similar in its pattern to expression observed in both endometrial and endometriosis-associated ovarian cancers. The components of the epithelial stem cells niche in endometriosis are shown in Figure 1.

## 4. Stem Cells in Endometrial Cancer

Stem cells are difficult to identify in the tissue of high proliferative and regeneration potential as endometrium. It is also not an easy task in endometrial cancer (EC). Many different molecules have been studied as markers of stemness in EC. One of them is CD117 (c-KIT) which upon SCF stimulation was shown to have significant proliferative activity and colony-forming capacity in EC. Other markers of EC stem cells are CD44 and CD55. CD44 is an adhesion molecule engaged in migration and metastases of EC, and is present on the cell surface of the spheres composed of CD44^+^CD133^+^ stem-like EC cells. CD55 is a cell surface complement inhibitor, and its high expression on the EC cells characterized the population of self-renewable and chemo-resistant stem-like cells [120,121]. Another marker, CD133 also called prominin-1, plays a role in the organization of the cell membrane [120]. 

EC CD133^+^ cells possessed more aggressive behavior and showed colony-forming ability and chemo-resistance. The side population (SP) of EC cells co-expressing ATP-binding cassette (ABC) transporters (ABCG2) and CD133 was shown to have a high proliferation rate and stemness features [122]. Endometrial cancer SP cells were also able to undergo EMT transition [48]. Signaling pathways playing a relevant role for CSCs function have been identified in EC stem cells. Regulators of NOTCH signaling pathway, MSI-1 protein, as well as expression of OCT-4 and SOX-2 transcription factors were found to be up-regulated or dysregulated in EC stem cells [123]. SOX2 is responsible for maintaining stem cell properties and differentiation restriction and was found to be upregulated only in low-grade EC [123]. Wnt/β-catenin pathway which activation promotes the proliferation and migration of cells was also up-regulated in EC CSCs [124]. Mutation in this pathway is generally regarded as a primary driver of carcinogenesis. Another signaling pathway engaged in the promotion of stemness in EC cells is the Hedgehog pathway. The increased expression of components of this pathway was noted in EC CSCs. Its aberrant activation leads to nuclei accumulation in β-catenin [125].

## 5. Stem Cells in Ovarian Cancer

Several surface cell markers identifying ovarian cancer stem cells (OCSCs) isolated either from patient samples or experimental animals and cancer cell lines have been described. Molecule CD44 is a cell-surface glycoprotein that is a receptor for hyaluronic acid receptor. The population of CD44^+^ OC cells possesses self-renewal, tumor-initiating and sphere-forming capacities. Recurrent OC shows higher expression of CD44-positive cells compared to primary tumors which is correlated with poor prognosis [126]. CD44 exists in alternatively spliced variants. Between them, CD44v6 was found in excess on OCSCs from distant metastases indicating metastasis-initiating activity. In patients with FIGO stage I-III OC distant metastasis-free survival was better in patients with CD44v6-low tumors [127]. CD117 recognizes a population of sphere-forming non-adherent OC identified with the “side population” of cells. The presence of CD117^+^ OC cells correlated with resistance to standard chemotherapy and shorter recurrence intervals in treated patients [128,129]. Double-positive CD44^+^/CD117^+^ cells are highly capable to recapitulate the original tumor after being transplanted into experimental animals [130]. It was found that CD133 mediates metastatic homing of ovarian cancer implants into the peritoneal tissue. Expression of intracellular stemness markers OCT4 and SOX2 is higher in CD133^+^ compared to the CD133^−^ cells [131]. The correlation between CD133 expression and advanced clinical stage, presence of ascites, and tumor non-responsiveness to chemotherapy, as well as patients’ survival, has been observed [132]. EpCAM (^+^) OC cells have greater tumor-initiating potential compared to EpCAM (−) cells, and EpCAM expression is increased in chemo-resistant tumors and correlates with unfavorable outcomes [133]. Another group of stemness markers is enzymes and intracellular molecules and transcription factors. Aldehyde dehydrogenase-1 (ALDH1)-positive cell phenotype identifies OCSCs population possessing self-renewal and stemness properties, and being capable of sphere formation and restoring the tumor. ALDH-1^+^ cells were found in both serous and CCOC ovarian cancers and were related to the worse survival of patients. Tumors exhibiting low expression of CD44^+^ALDH-1^+^ cells showed a better response to chemotherapy and longer progression-free survival [134]. Recurrent platinum-resistant ovarian tumors compared to primary tumors are enriched in the population of CD44^+^CD133^+^ALDH1A1^+^ OCSCs. Similarly, CD44^+^/CD24^+^/EpCAM^+^ cells show OCSCs properties having increased migratory and invasive potential and chemo-resistance [135]. Musashi-1 and ALDH1 expression were significantly higher in HGSOC, mucinous adenocarcinomas and CCOC compared to benign tumors and normal tissues, as well as in advanced and lymph node metastatic tumors compared to early stage lymph node-negative tumors [136]. Overexpression of MSI-1 alone is also associated with an unfavorable prognosis in OC patients. Inhibition of MSI-1 function reverses chemo-resistance and promotes apoptosis of cancer cells [137]. Expression of NANOG in OCSCs cells correlates with clinical stage and high grade, as well as resistance to standard chemotherapy [138]. Over-expression of SOX2 is related to the stemness of cells via up-regulation of resistance to apoptosis. In OC SOX2-positive cells were identified in the tubal epithelium of patients with high-grade OC tumors of poor outcome and in patients with germline BRCA1/2 mutations [139,140]. Similarly, up-regulation of OCT4 in OCSCs was correlated to tumor progression and chemo-resistance [141]. NANOG, OCT4 and SOX2 were over-expressed both in tumor tissues and in cellular spheres built from OCSCs cells circulating inside ascites [127,142]. The higher expression of SUSD2 in HGSOC ovarian cancer was correlated with worse overall survival, recurrence, platinum chemoresistance and lymph node metastases. Over-expression of SUSD2 promotes EMT and metastatic capacity of cancer cells through the regulation of EpCAM [143]. SUSD2 is one of the Notch3 downstream genes, while high SUSD2 expression is associated with the OC progression. In Table 1 similarities between stem cells in endometriosis and ovarian cancer have been shown.

## 6. ARID1A/PI3K/AKT Pathway in Endometriosis and EAOC

ARID1A is considered a cancer-inhibiting gene that encodes ARID1A protein belonging to the chromatin remodeling complex. It plays an important role in carcinogenesis as a tumor suppressor. Mutation of ARID1A gene results in reduced expression and dysfunction of ARID1A protein followed by a change in expression of several genes regulating the proliferation of cells and change in the activity of PI3K/AKT signaling pathway. Mutations in the ARID1A gene have been found in various cancers, but are particularly frequent in CCOC and ENOC, as well as in endometroid and clear-cell endometrial cancers [154,155]. Activation of PI3K/AKT pathway regulates cell proliferation, adhesion and resistance to apoptosis thus increasing the growth and survival of cancer cells [156]. 

### 6.1. ARID1A/PI3K/AKT Pathway in Endometriosis

Mutation of the ARID1A gene was demonstrated in atypical endometriosis originating from ovarian endometriomas and localized adjacent to ovarian CCOC and ENOC tumors. Mutations were not present in remote non-atypical endometriotic lesions of the same patients [136]. However, there is still controversy existing at which stage of endometriosis ARID1A mutation does occur. Immunohistochemical studies proved a high correlation between ARID1A gene mutation and loss of ARID1A protein expression in the studied tissue. Therefore, although the sequencing studies in non-atypical endometriosis are lacking, the IHC studies could serve as a good approximation of mutational ARID1A gene status. These studies indicated that loss of ARID1A expression could be shown also in the cases of non-atypical endometriosis in ovarian endometriomas. The study of Yamamoto et al. showed that 86% of tumor-associated non-atypical endometriosis and 100% of atypical endometriosis were ARID1A-deficient [157]. Samartzis et al. investigated samples of non-atypical ovarian and deep-infiltrating endometriosis (DIE) showing the lack of ARID1A expression in 15% and 5% of endometriomas and DIE, respectively [158]. In Xiao et al. study the loss of ARID1A expression was observed in 20% of non-atypical endometriomas [159]. Moreover, as was noticed earlier, ectopic stem cells from endometriotic non-atypical lesions showed reduced expression of ARID1A [160]. ARID1A low expression is observed in regenerating tissues and enhances tissue repair processes. Inactivation or epigenetic silencing of the ARID1A gene in the epithelial component of endometriosis may result from chronic inflammation and cyclic regeneration [161,162]. Activation of PI3K/AKT pathway was described in endometriosis [163]. PI3K/AKT pathway regulates inside endometrium expression of forehead-box O family FOXO1 protein and insulin growth factor binding protein-1 (IGFBP-1) which both are engaged in the decidualization of the endometrium. Increased activation of PI3K/AKT pathway causes reduced decidualization both in endometriotic lesions and in eutopic endometrium of patients with endometriosis [164]. This mechanism may be responsible for reduced decidualization in response to progestins in endometriosis [165]. 

### 6.2. ARID1A/PI3K/AKT Pathway in EAOC

Somatic mutations of the ARID1A gene was observed in 46–57% of CCOC and in 30–48% of ENOC, respectively [166,167]. Most of the mutations correlated with a loss of ARID1A expression in IHC—73% of mutated CCOC tumors and 50% of mutated ENOC tumors, respectively [166]. Mutations of the ARID1A gene are regarded as the main genetic disturbances present in CCOC and ENOC cancers [168]. 

Activation of PI3K/AKT pathway is a quite common event in CCOC and ENOC, and in the case of CCOC it occurs in 14–40% of tumors dependent on the type of activating mutations, which could originate from AKT2 amplification, loss of PTEN expression or activation of catalytic p110α subunit of PI3K [169,170]. Overactivity of PI3K/AKT pathway in ovarian cancer could augment chemo-resistance, especially in CCOC tumors [171]. The coexistence of PIK3CA mutations that encodes p110α subunit and loss of ARID1A expression was found in 47% of ARID1A-deficient CCOC tumors. An association between PIK3CA and ARID1A mutations was also reported for endometrial endometrioid cancer [158]. Animal studies indicate that ovarian carcinogenesis may demand both mutations to occur [172]. In this context, the loss of ARID1A expression in atypical endometriosis might be an initial but not sufficient step in ovarian carcinogenesis, which starts after the second mutation event concerning the PI3K/AKT/PTEN pathway [173].

## 7. Other Gene Mutations in Endometriosis

The mutations of genes in endometriosis could result from the natural history of endometriotic lesions in which progressive fibrogenesis occurs which may influence gene expression. Fibrogenesis is a pathological process of deposition of excessive amounts of extracellular matrix (ECM) components resulting from uncontrolled tissue repair. There is no doubt that cyclic bleeding and subsequent tissue repair do occur in endometriotic lesions and ovarian endometriomas leading to “repeated injury and tissue repair” (ReTIAR) syndrome [174,175,176]. Fibrotic tissue is present in the endometrioma capsule, in peripheral parts of peritoneal implants and DIE. The sequence of events may start from platelet activation, followed by platelet-derived TGF-β secretion, which initiates smooth muscle metaplasia (SMM) and fibrosis, through EMT and fibroblast-to-myofibroblast differentiation (FMT). The pro-inflammatory environment inside the peritoneal cavity could modulate this sequence [177]. The fibrogenesis in the liver in the course of ReTIAR syndrome following viral hepatitis is characterized by similar changes mediated by TGF-β, growth factors and pro-inflammatory cytokines [178]. Moreover, in both diseases inflammation and fibrosis cause hypoxia which promotes angiogenesis. The similarities go even further, as both in chronic hepatitis and in endometriosis (especially endometriomas) iron overload generates ROS, inflammation and oxidative stress which induce fibrosis [179]. Finally, it is known, that fibrogenesis is one of the strongest risk factors for hepatocellular cancer [178]. The question comes up if fibrogenesis and fibrogenesis-mediated changes of gene expression could be a risk factor for the progression of endometriosis into EAOC. The disturbances in the expression of the markers of DNA damage like 8-hydroxydeoxyguanosine (8-Oh-dG) and histone protein γ-H2AX which were found in the proximity or inside the endometriotic lesions seem to support this suggestion [180,181]. The mutational pressure of oxidative stress, iron overload and hypoxia on endometriotic lesions could be dependent on the status of the lesion and differ in non-fibrotic and fibrotic lesions [180]. Moreover, endometriotic implants show an increased level of DNA damage and decreased DNA repair activity in the course of the disease [182]. Mismatch repair (MMR) activity differs in stromal and epithelial components of the lesions, being lower in stromal and higher in epithelial cells, respectively [183]. Microsatellite instability (MSI) associated with the inactivation of MMR protein MLH1 has been described in the malignant transformation of endometriosis [184]. Loss of MMR activity was noted in 10% of EAOC tumors [185].

TP53 is a tumor suppressor gene frequently mutated in several malignancies. The results on TP53 mutation in endometriotic lesions are conflicting, as both over-expression and down-regulation of TP53 were reported [186,187,188]. However, based on more detailed genetic studies [189,190] there is a rational supposition that TP53 is down-regulated or silenced in endometriotic lesions. Loss of TP53 activity leads to increased fibroblast activation, decreased immune surveillance, ECM deposition and fibrosis. Micro RNA miR-125b inhibits TP53 expression and promotes fibrogenesis, and miR-125b over-expression was found in endometriosis [191,192]. Similarly, activation of the Wnt/β-catenin pathway and inactivation of peroxisome proliferator-activated receptor gamma (PPARγ) observed in endometriotic lesions could further augment fibrosis [193]. Inactivation of TP53 observed in endometriotic stromal cells may not be univocal with pre-malignancy, however, could influence carcinogenesis in favorable conditions [194]. 

PTEN is another tumor suppressor gene. PTEN mutations have been recognized in 21% of endometriomas and half of advanced rASRM III/IV grade lesions. Loss of PTEN expression was also reported in endometriosis malignant transformation [195,196,197]. Inhibition of PTEN and activation of the PI3K/AKT pathway enhances proliferation and reduces apoptosis in endometriotic stromal cells and fibroblast [198,199,200]. In fibrotic diseases, PTEN expression is decreased or absent [201]. Some modulators of PTEN expression, like miR-21 or CTGF were found to be up-regulated in endometriosis [202]. Enhancer of zest homolog-2 (EZH2) participates in histone methylation and mediates transcriptional repression [203]. EZH2 expression is elevated in endometriosis and it functions as a stimulator of EMT and inducer of PTEN inhibition leading to enhanced fibrosis in endometriotic lesions [204,205]. 

KRAS is an oncogenic protein encoded by KRAS-2 gene and engaged in the EGFR signaling pathway [206]. KRAS can be activated by many factors including TGF-β, EGF, PDGF which are activated in endometriosis [207,208]. Elevated KRAS expression was described in the eutopic endometrium of patients with endometriosis [209]. KRAS mutations were also described in 29% of ENOC tumors [210]. KRAS protein is capable to activate downstream signaling via the ERK pathway. ERK activation is crucial for EMT and fibrogenesis. TGF-β can stimulate ERK/MAPK/JNK signaling pathway to induce EMT and fibrogenesis. The members of this pathway were all found to be engaged in the pathogenesis of endometriosis. KRAS activates also SCF which binds to c-KIT/CD117 receptor and activates ERK and PI3K-dependent pathways. Increased concentrations of SCF as well as c-KIT were described in peritoneal fluid and ectopic implants of endometriosis patients [85,211].

NOTCH1 is a protein responsible for the signaling pathway regulating cell proliferation, differentiation, apoptosis and cell stemness or quiescence [212]. Protein is encoded by gene NOTCH1 which due to the multifunctional role of the protein is viewed as both like oncogene and cancer suppressor [213]. NOTCH1 was found to be elevated in localizations adjacent to peritoneal endometriosis implants but decreased in eutopic endometrium in endometriosis patients [214]. Increased expression of MSI-1 which acts as a positive regulator of NOTCH1 was reported in endometriosis [35]. Elevated NOTCH1 has been found to stimulate renal fibrosis, while loss of NOTCH1 activity has decreased pulmonary fibrosis [215]. Oxidative stress in endometriosis activates NOTCH1 signaling and promotes fibrosis in implants [216]. 

GATA-binding protein-2 and -6 are transcriptional factors and key regulators of steroid hormones receptor expression in both eutopic and ectopic endometrium. The epigenetic regulation (DNA methylation) of genes GATA2 and GATA6 has been noticed in endometriosis. The GATA2 was found to be unmethylated and abundant in stromal cells of eutopic endometrium, but methylated and inactive in stromal cells of endometriosis. On the opposite, GATA6 was found to be methylated and inactive in eutopic endometrium stromal cells, while was active and unmethylated in endometriosis stroma [217]. GATA2 is expressed in healthy endometrial cells and up-regulates genes engaged in endometrial decidualization. Ectopic expression of GATA6 prevents decidualization of ectopic endometrium and pushes it toward pathological phenotype [218]. Depletion of GATA2 decreases the expression of progesterone receptor (PR) in the endometrial epithelium in mice, and in humans reducing the markers of decidualization. Over-expression of GATA6 in endometriosis resulted in changes in the expression of hormone receptors—reduction of ERα and PR, and stimulation of ERβ [3]. Disturbed ERβ/ERα ratio in endometriotic stromal cells prevents induction of the PGR gene (gene for PR receptor). It also stimulates cell proliferation and survival. Through interactions with IL-1β and steroid receptor coactivator-1 (SRC1), ERβ prevents cell apoptosis and enhances inflammatory reaction [219]. In endometrial stromal cell cultures, GATA6 alone is essential but not sufficient for estrogen formation and needs cooperation with the nuclear receptor subfamily 5, group A, member 1 (NR5A1). The presence of both GATA6 and NR5A1 is required for estradiol production, which is critical to the transformation of stromal cells into endometriotic-like cells [217]. High expression of GATA2 was correlated with better survival, while high expression of GATA6 was an unfavorable prognostic factor in OC patients [220]. 

The next-generation sequencing technology applied to genetic profile evaluation of endometriosis and ENOC indicated that several mutated genes were common for both types of pathology and that similar pathways were altered in both endometriosis and ENOC. However, there were also identified mutated genes that were characteristic for each group: JAK3, KRAS and RB1 for endometriosis; and ATM, BRAF, CDH1, EGFR, NRAS, RET and SMO for ovarian ENOC. There were also genes methylated in endometriosis, mainly PYCARD, RARB, RB1, IL2, CFTR, CD44 and CDH13; and ENOC—MLH3, BRCA1, CADM1, PAH [221]. 

## 8. Galectins in Endometriosis and EAOC

Galectins are glycan binding proteins specifically binding to β-galactoside sugars. They contribute to intercellular interactions, extracellular matrix-cell interactions, apoptosis, migration, angiogenesis and inflammation [222]. Among several known galectins, there are galectin-1, -3 and -9 which are engaged in both progression of endometriosis and OCs [222]. Galectin-1 is expressed in both epithelium and stromal component of the endometrium, while galectins-3 and -9 are present in epithelium and decidua, but not in the stroma [223]. Galectin-1 was shown to be over-expressed both in the endometriotic lesions and in eutopic endometrium of patients with endometriosis. Galectin-1 facilitated inflammation and angiogenesis in ectopic endometrium. Monoclonal antibodies against galectin-1 were able to slow the growth of endometriotic implants [223]. 

Analogically to galectin-1, galectin-3 expression was enhanced in endometriosis and eutopic endometrium of endometriosis patients [224]. Loss of galectin-3 activity resulted in down-regulation of VEGF, TGF-β and cyclooxygenase-2 (COX-2) in endometriotic lesions [225]. Galectin-3 was shown to interact with KRAS upon epidermal growth factor (EGF) stimulation, followed by the stabilization of the KRAS-GTP complex, which regulates cell proliferation and inhibits apoptosis [226]. Increased concentrations of galectin-9 were found in the serum of patients with endometriosis [227]. 

Galectins modulate the pathways controlled by oncogenes and tumor suppressor genes, therefore their aberrant expression could facilitate carcinogenesis, especially in the environment characterized by chronic inflammation like in the case of endometriosis [228,229]. The interaction of galectin-1 with extracellular matrix components could promote cancer survival and migration [230]. In EAOC the expression of galectin-1 was found to be increased, while high expression in the stromal tumor compartment was a poor prognostic marker. In CCOC the higher expression of galectin-3 was associated with increased tumor invasiveness, through the up-regulation of the NF-κB signaling pathway. Galectin-3 expression is correlated also with fast-growing, advanced and chemo-resistant tumors [231]. Galectin-9 functions as a modulator of the anti-tumor immune response. Through interaction with receptors on the surface of T cells, galectin-9 could induce their apoptosis and enhance the tolerance toward tumor cells [228].

## 9. Chaperones in Endometriosis and Cancer Stem Cells

Chaperones are a class of proteins that regulate the folding of polypeptide chains and conformational changes of so-called client proteins [232]. The heat shock proteins (HSPs), glucose-regulated proteins (GRPs), TNF receptor-associated protein-1 (TRAP1), calreticulin and others belong to this class of molecules [233,234,235]. Chaperones HSP70 and HSP90 stabilize cytosolic proteins, prevent their aggregation, stimulate their degradation, regulate their maturation and modulate intracellular signal transduction [236]. GRPs and calreticulin are responsible for the regulation of transport and processing of proteins inside the endoplasmic reticulum, while TRAP1 functions similarly inside mitochondria [235,237]. Inducible forms of HSPs are a part of the “heat stress response” resulting from different stressors like acidosis, hypoxia, ROS, or environmental toxicity. This reaction is mediated by heat shock transcription factor-1 (HSF1) [238]. Chaperones function also in cancer cells inducing their resistance to a hostile environment and toxic therapy, and the up-regulated levels of some HSPs were observed in several malignancies [236]. Among the HSPs client proteins are regulators of cell stemness which contribute to the renewal and survival of cancer stem cells (CSCs). The client proteins for HSP90 include: survivin, HIF-1α, metalloproteinases MMP2 and MMP9, EGFR and AKT [239,240]. Elimination of HSP90 activity in tumor cells inhibited their migration, invasiveness and metastatic potential, while activation of the NANOG-dependent HSP90/TCLA1/AKT signaling pathway augmented significantly the stemness of tumor cells [241,242]. HSP90 function was also important for AKT/ERK/JAK/STAT3 signaling pathway responsible for the generation of CD44^+^CD24-ALDH1^+^ CSCs phenotype [243]. Another HSP protein, HSP70 is the key element of the HSF1-mediated response to stress. HSP70 is capable to diminish the adverse effects of stress on the cell, and to inhibit pro-apoptotic pathways in the cell subjected to stress [236]. Tumors indicate up-regulation of HSP70, which is correlated with their aggressiveness and chemo-resistance. Enhanced expression of HSP70 was found in cells showing stemness markers and high metastatic potential, while inactivation of HSP70 resulted in a decrease of CSCs and impairment of both tumor invasion and metastases formation [244]. In ovarian cancer HSP70 knockdown down-regulated the EMT- and stemness-associated proteins [245]. Activation of HSF1 also could support stemness and population of CSCs. Triggers of HSF1 activation, like hypoxia, acidosis, and inflammation are also activating stimuli for EMT and acquirement of CSC phenotype. HSF1 was shown to induce tumorigenesis and metastases in mice model of cancer, as well as in human ovarian tumors [246,247]. In breast cancer, the HSF1 expression was correlated to stemness markers expression and chemo-resistance [248]. Stress-induced phosphoprotein-1 (STIP-1) coordinates functions of HSP90 and HSP70 in protein folding. Increased expression of STIP1 was described in ovarian and endometrial cancer [249]. Both epithelial and stromal cells of endometriotic lesions showed expression of STIP1, and serum levels of STIP1 were higher in endometriosis patients compared to controls [250]. It was also shown that HSP70 induced pelvic inflammation and was probably involved in the growth of endometriotic implants [251]. Moreover, the HSF1 expression was increased in endometriosis and promoted endometriosis development through the enhancement of glycolysis in endometriotic cells [252]. Endometriosis cells, similarly to tumor cells, use preferably aerobic glycolysis to get energy. This type of glycolysis in cancer promotes angiogenesis, cell invasion and tumorigenesis, while endometriosis activates survival signals [253]. Chaperons are also represented by small HSPs (HSP27, HSP20, alpha B-crystallin) which prevent the mistaken folding of proteins, inhibit apoptosis, induce proliferation and metastases in cancer [254,255,256]. Elevated concentrations of small HSPs were found in exosomes, serum and peritoneal fluid of patients with ovarian and endometrial cancer, but also endometriosis [257]. 

## 10. miRNA and Endometriosis

Micro RNAs (miRNA) are a population of small non-coding RNAs that function as regulators of gene expression through changes in gene translation and post-translational modification of mRNA stability [258]. One of the most discussed miRNAs in the pathogenesis of many diseases is the miR-200 family. Down-regulation of miR-200 family RNAs induces an EMT transition in several cancers including endometrial cancer [259]. MiR-200 was also found to be down-regulated in ectopic endometrium [260]. Another miRNA, miR-199a through its down-regulation, targets and enhances the expression of IL-8 and NF-κB molecules. Down-regulation of miR-199a was confirmed in ectopic endometrium and ovarian endometriomas and may be responsible for increased invasive potential and infiltration of the implants [261]. Cytokines TGF-β and IL-8 are engaged in inflammation and tissue repair in endometriosis. MiR-20a targets regulatory signals for TGF-β and IL-8 secretion and its down-regulation increases concentrations of these cytokines and the growth of endometriotic implants. The next miR-143 molecule was found in higher concentrations in the serum of endometriosis patients. Up-regulation of miR-143 inhibits the expression of fibronectin type III domain containing 3B (FNDC3B) and stimulates invasion and migration of endometriotic cells [262]. Siruin-1 (SIRT1) is the regulator of chromatin remodeling and cellular signaling via its action on histones, p53, forehead box O (FOXO) and NF-κB proteins [263]. MiR-34a functions as one of the regulators of SIRT1 expression, and over-expression of miR-34a decreases SIRT1 level. Mir-34a together with p53 and SIRT1 participate in the miR-34a/p53/SIRT1 pathway having important regulatory properties in endometriosis and cancer [264,265]. In endometrial lesions miR-34a, p53, pro-apoptotic Bax, Bcl-2, and FOXO-1 proteins were found to be down-regulated, while SIRT1 and anti-apoptotic Bcl-xL proteins were up-regulated. This observation supports the notion that the miR-34a/p53/SIRT1 pathway plays a role in decreasing cell apoptosis in endometrial lesions [266]. The regulation of SIRT1 by miR-34a could also influence angiogenesis in endometriotic implants [267]. The enhanced expression of another miR-125b was noticed in both endometriosis and cancer. MiR-125b plays important role in the regulation of the proliferation and migration of cells [268,269]. A negative correlation between miR-125b and TP53 expression was found in endometriosis. While miR-125b was significantly over-expressed in ectopic endometrium, there was a decrease of TP53 expression observed, both in ectopic and eutopic endometrium of endometriosis patients. This observation implies a possible role of the miR-125b/TP53 pathway in the pathogenesis of endometriosis and its pro-cancerous potential [270]. The next important miRNAs in the pathogenesis of endometriosis are the let-7 family members. Let-7 family miRNAs regulate cell differentiation on many different levels and are considered as tumor suppressors, because of the negative regulation of Ras oncogenes and loss of activity observed in many cancers [271]. Low levels of the let-7 expression resulted in KRAS activation and progesterone resistance in severe endometriosis [272,273]. 

Many studies have shown several both up-regulated and down-regulated miRNAs in the serum of endometriosis patients. The panel of miRNAs was different in low-grade early (I/II) and high-grade advanced (III/IV) endometriosis, respectively. Among up-regulated miRNAs in early endometriosis were: miR-185, miR-242, miR-296, miR-424, miR-502, miR-542, miR-550 and miR-636 [274]. In advanced endometriosis up-regulated miRNAs were as follows: miR-18a, miR-125b, miR-342, miR-500a and miR-451a [268]. Down-regulated miRNAs in advanced endometriosis were: miR-34c, miR-9, let-7b, miR-125a, miR-3613, and miR-6755 [268,275]. These serum miRNAs were studied as potential biomarkers of endometriosis and their meaning for endometriosis growth and possible malignant transformation has not been studied precisely. The serum panel of miRNAs described in the serum of CCOC and serous OC patients was shown to differ from the miRNAs panel described in endometriosis, but again the importance of this observation for malignant transformation of endometriosis into EAOC is uncertain [276,277]. The profiling studies in endometriotic tissues have given more stable results. More than one research indicated disturbed expression of the following miRNAs: miR-1, miR-29c, miR-34c, miR-100, miR-141, miR-145, miR-183, miR-196, miR-200 a-c, miR-202, miR-365 [278]. Most of them are known regulators of angiogenesis, cell proliferation, adhesion and invasion, as well as EMT transition. The most frequently identified targets for disturbed miRNAs in endometriosis were signaling pathways engaging estrogen and progesterone receptors, homeobox protein transcription factor HOX-A10, c-Jun, Wnt/β-catenin, AKT and cyclin D1 [279,280]. Other targets identified were VEGF, metalloproteinases MMP-3 and MMP-9, tissue inhibitors of metalloproteinases (TIMP), and regulators of EMT like ZEB1 and TGF-β1 [281]. In ovarian cancer members of the miR-200 family (miR-141, miR-200a, miR-200b, miR-200c) and miR-199a, miR-140, miR-145, miR-125b were dysregulated similarly as in the case of endometriosis [282]. 

## 11. Transformation of Endometriosis into EAOC

The mutations of several genes were found in all types of endometriosis including iatrogenic incisional endometriosis. Between them, somatic mutations in cancer-driver genes like KRAS, PI3KCA, PTEN, and TP53 were noted [283,284]. These mutations were present in the epithelium of endometriotic lesions. What is interesting, some of these mutations have been also observed in normal eutopic endometrium [285,286,287], meaning that mutations in regenerating cyclically tissue like endometrium or endometriosis may be an evolutional advantage allowing for fast renewal of this tissue. Therefore, mutations are probably not associated directly with carcinogenesis. From another point of view, the frequency of mutations in endometrium increases along with the progression of women’s age, and the vast majority of endometrial cancers are recognized in peri- and postmenopausal women. It is possible that other factors affecting the pre-mutated cells could play a role as a trigger mechanism for carcinogenesis [288]. They could consist of the “second-hit” mutation of DNA polymerase epsilon (POLE) or MMRd genes [288]. Alternatively, they could originate from a unique environment existing inside endometriotic lesions, mediated by stromal cells. In contrast to epithelial cells, a stromal component of endometriotic lesions lacks any cancer-driver mutations. However, stromal cells show epigenetic defects influencing estrogen regulation, progesterone deficiency and the creation of a pro-inflammatory environment [289]. The features of the endometriotic microenvironment are: hypoxia, inflammation, extracellular matrix, changed metabolism, and steroid hormones. Their orchestrated action could push the endometriotic epithelial cells toward cancer. Hypoxia in endometriotic lesions stabilizes HIF-1***α***, which enhances proliferation and angiogenesis. The IHC studies showed a correlation between HIF-1***α*** expression in endometriosis precursor lesions and matched CCOC. Hypoxia and inflammation accompanying the cyclic menstrual changes in endometriosis stimulate the tissue factor (TF) which could trigger hyper-coagulation [290]. The increased risk for thromboembolism is a typical clinical symptom of CCOC. Fibroblasts and ECM components play important role in endometriosis. Endometriotic ECM components mediate signaling between epithelial and stromal cells [291]. Fibroblasts in endometriosis indicate an over-expressed ERK signaling pathway, progesterone resistance increasing their proliferation and pro-inflammatory phenotype [292,293]. Similarly, in ovarian cancer ECM components and cancer-associated fibroblasts (CAFs) constitute the indispensable components of the cancer stem cells niche [37]. Immune cells and cytokines are very important components of the endometriotic environment. Macrophages are the necessary stimulator of endometriotic lesion growth, and their recruitment toward the lesions is mediated by hypoxia, iron overload and inflammation. Cytokines TGF-***β***, TNF-***α*** and IL6, IL8 and monocyte chemotactic protein-1 (MCP-1) play a key role in inflammation inside implants and peritoneal fluid [220,294]. In ovarian cancer tumor-associated macrophages and inflammatory environment in the cancer niche also enhance the proliferative abilities of CSCs [37]. Ovarian endometrioma contains hemolyzed blood with an excess of iron and heme products which further contribute to chronic inflammation and produce fibrosis, oxidative stress and predisposes to gene mutations [295,296]. Therefore, iron overload in endometriosis may promote highly mutated cells possessing higher survival rates [297]. Elevated lactate levels and acidosis resulting from glycolysis in ectopic endometrium promote cell survival [298]. The analogical mechanism supports stemness in CSCs niche in ovarian cancer. Endometriotic lesions indicate high expression of aromatase (CYP19A1) which converts androstenedione and testosterone to estrone and estradiol. High levels of estradiol promote inflammation. Aromatase is also stimulated by prostaglandin E2 [295,299]. The role of estradiol in EAOC cancer biology needs further studies, but its proliferative activity could account for malignant transformation. MiRNAs are a frequent target for dysregulation in both endometriosis and EAOC. One of the miRNAs with disturbed expression was miR-126, which down-regulation was observed in ectopic compared to eutopic endometrium, as well as in CCOC and ENOC, where its decreased expression was a predictor of poor outcome [300,301]. Some other miRNAs were observed in endometriosis and ovarian cancer in general. MiR-135 was found to be down-regulated both in endometriotic lesions and ovarian cancer, which correlated with increased expression of HOXA10 and worse survival [279,302]. Over-expression of miR-325 and miR-492 was also observed in endometriosis and ovarian cancer. The first one is engaged in the regulation of autophagy in hypoxic conditions, while miR-492 decreases PTEN expression [303,304]. Besides miRNAs, other mechanisms of epigenetic regulation, like DNA methylation, have been probably engaged in the malignant transformation of endometriosis. Promoter hypermethylation followed by inactivation of Runt-related transcription factor 3 (RUNX3) was noted in 60% of patients with EAOC, and was also present in eutopic endometrium of EAOC patients [305]. Progressive hypermethylation of RUNX3 could be implicated in the malignant transformation of ovarian endometriosis. Similarly, the absence of MLH1 expression resulting from the hypermethylation of the gene promoter was associated with the malignant transformation of ovarian endometriosis [306]. The methylation of genes engaged in the estrogen ER-***α*** receptor signaling pathway was connected with the progression from endometriosis to CCOC [307]. 

## 12. Environmental Risk Factors—Endometriosis and Ovarian Cancer

Epidemiological studies revealed several risk factors associated with the development of endometriosis. Toxins defined as endocrine disrupting chemicals (EDCs) are well known factors causing harmful effects on female reproductive organs. Dioxin (TCDD-2, 3, 7, 8-tetrachlorodibenzo-p-dioxin), polychlorinated bisphenol (PCBs) and phthalates belong to this category of toxins. Dioxin is a by-product of plastic and fuel burning and in the production of plastic compounds and pesticides. Its presence was confirmed both in the air and water [308]. PCBs are used in rubber, adhesives and paint industry and besides circulation in the environment can concentrate in adipose tissue [309]. Phthalates are frequently used in plastic, toy and cosmetic industry, and are found in both serum and urine of humans exposed to them. The animal Rhesus monkey model indicated, that contamination with the above-mentioned TCDD provoked the development of endometriosis dependent on the concentration of chemicals [310]. In mice and rat model, exposure to TCDD resulted in a significant increase in the diameter of endometriotic lesions [311]. Phthalates have a proliferative effect on endometrial tissue, and patients with endometriosis were shown to have increased concentrations of phthalates in both serum and urine [312]. 

Other environmental risk factors considered in endometriosis are alcohol consumption, as well as smoking. Alcohol use is a risk factor for endometriosis, and the odds ratio for development of the endometriosis in moderate and heavy drinkers were 1.7 and 1.8, respectively [313]. The possible mechanism underlying that connection is an increase in E2 concentration and impairment of immunity resulting from alcohol consumption [314,315]. Smoking has probably both beneficial and negative effects on endometriosis. Smoking alters the metabolism of E2 and inactivates estrogens, thus protecting against endometriosis. However, from another point of view, dioxins present in cigarette smoke could have adverse effects on the growth of endometriosis [316]. Another problem is a possible association between diet habits and the risk of endometriosis [317]. It was shown, that a higher intake of citrus fruits was correlated with a lower risk of endometriosis [318]. Similarly, a diet containing low nickel concentrations was able to reduce gastrointestinal and gynecological symptoms in endometriosis patients [319]. The results of the prospective cohort study indicated that red meat intake was associated with a 56% higher risk of endometriosis [320]. The connection between red meat and endometriosis may be based on the observation that meat and fat consumption influences the sex hormone metabolism and increases the heme iron intake. The latter connection may be explained by an increase in inflammation and oxidative stress mediated by heme iron in meat consumers [321,322]. Moreover, the meat-rich diet contains omega-6 fatty acids which are responsible for the enhancement of estrogen synthesis, inflammation and promotion of endometriosis [323]. Women with endometriosis seem also to consume fewer vegetables, dairy products and omega-3 unsaturated fatty acids [324,325]. 

In ovarian cancer, a high-fat diet could promote carcinogenesis by stimulating the effect of estrogen synthesis [326]. The same effects were associated with a pro-inflammatory diet rich in saturated fatty acids and sugar [327]. Omega-3 unsaturated fatty acids in the diet seem to decrease endogenous estrogen synthesis and reduce the risk of ovarian cancer [323]. The higher consumption of meat was connected with worse survival in OC patients, conversely to the vegetables and fruits which seemed to prolong survival [328,329,330]. However, the systematic review of papers devoted to dietary habits and OC risk failed to produce conclusive recommendations [331]. 

In endometrial cancer, a higher fat intake and a sugar-rich diet were both risk factors (OR 1.72 and 1.84, respectively) [332,333]. Meta-analysis revealed that consumption of omega-6 fatty acids and meat is also a risk factor for endometrial cancer in obese women [334]. 

## 13. Stem Cells and Environmental Risk Factors

Exposure to environmental toxins could result in the accumulation of DNA damage and epigenetic alterations [335]. These changes could be particularly dangerous for the population of somatic stem cells and could have a potential role in malignant transformation. Stem cells possess increased capability to repair DNA, however recurrent or prolonged exposition to environmental toxins could exhaust DNA repair possibilities or could affect epigenetic regulation followed by the changes in gene expression [336]. Even subtle toxin-mediated epigenetic changes could result in profound changes in the function of transcriptome if they occur in the periods of the individual increased susceptibility called “critical windows” [337]. These may influence the individual predisposition to malignant diseases or could be passed to the next generation changing children’s gene expression. Epigenetic transgenerational inheritance of adult-onset disease induced by bisphenol and phthalates was observed [338]. Dioxins were shown to bind to aryl hydrocarbon receptor AhR which functions as a transcription factor regulating the expression of genes responsible for cell growth, differentiation and drug metabolism [339]. Several toxins and heavy metals can modify DNA methylation and increase oxidative stress, which could alter gene expression in stem cells contributing to malignant transformation [340,341]. Pesticides were found to change the expression of miRNAs regulating Wnt/β-catenin and p53 pathways [342]. Besides pesticides there are heavy metals, organic pollutants and cigarette smoking that can alter miRNA function, also the improper or unbalanced diet could detrimentally affect stem cells. The high-fat diet was found to induce the proliferation and function of LGR-5^+^ intestinal epithelial stem cells and augment their capacity to create tumors [343]. A high-fat diet has also been shown to enhance tumor aggressiveness and enrichment in CSCs in a mouse model of glioblastoma [344], and to increase metastatic potential in oral squamous cancer [345]. Contrarily, an in vitro model of triple-negative breast cancer showed, that starvation (low serum, low glucose) reduced the proportion of CD44^+^ and ALDH-1^+^ stem cells, and in tumors transplanted to mice, it slowed down cancer growth and initiated tumor cell apoptosis [346]. Caloric restriction in the diet reduces CSCs renewal in breast cancer and affects CSCs function through PI3K/AKT/mTOR/S6kinase signaling-dependent mechanisms [347,348]. A proper diet could also influence the niche of CSCs and diminish the pro-inflammatory environment [349]. The connection between diet and CSCs function is based on the AMPK/mTOR/SIRT1 pathway. Low ATP levels in cells as a result of fasting activate AMP-activated protein kinase (AMPK) which modulates glucose and lipid metabolism, as well as mTOR pathway and SIRT function. This sequence of actions is followed by an increase in CSCs apoptosis, a decrease in proliferation, and disturbed response to hypoxia and autophagy [350,351,352,353]. 

## 14. Working Hypothesis

Endometriosis is a proliferative disease that in many aspects behaves similarly to neoplastic tumors. It can infiltrate tissues, migrate into distant locations, create its blood vessels, proliferate vigorously and avoid the effective immune response of the host. An explanation that this analogy is only a sign of the high regenerative capacity of endometriosis does not seem to be convincing, although a low rate of progression from endometriosis to EAOC (merely a few percent of cases) could support such a notion. The similarity of mechanisms ruling the development and progression of endometriosis and cancer is probably not accidental. Long-lasting and advanced endometriosis more frequently progresses into the atypical lesion and finally into ovarian cancer. The older age (>45 years) and larger size of endometrioma are also considered risk factors for the progression of endometriosis into EAOC [354]. These observations suggest that in non-treated patients the true risk of progression into a malignant tumor could be higher than the reported 1.5% during the lifetime. The reason for the low malignant transformation rate could be the fact, that patients with symptomatic endometriosis and infertility are diagnosed and treated laparoscopically and pharmacologically. Thus, the natural evolution of endometriosis is stopped by medical treatment, and only a few cases have enough time to progress toward EAOC. Another explanation is, that progression of endometriosis toward cancer is a multistep long-lasting process dependent on many risk factors. Genetic predisposition resulting from inherited polymorphism or mutations of several genes or alternatively from epigenetic transgeneration gene modification could be one of the main risk factors. Other ones are exposition to environmental toxins and improper nutritional habits. Finally, inflammatory local reaction, oxidative stress and mutational influence of heme products could account for the malignant transformation of endometriosis. We propose, that the main common target for the action of all these factors are endometriosis epithelial stem cells, and that stromal mesenchymal/bone marrow-derived stem cells are one of the most relevant modulators of carcinogenesis via CTGF secretion, and modulation of the endometriotic niche (shaping resistance to hypoxic stress, regulation of mesothelial-to-mesenchymal transition and fibrosis, ECM remodeling and angiogenesis). The endometriosis SCs could probably originate from epithelial stem cells of eutopic endometrium and tubal endometriosis, or alternatively from SP stem cells able to differentiate into both epithelial and stromal SCs cells. ARID1A gene mutation and activation of PI3K/AKT pathway may be an initial change in endometriosis SCs, followed by gene modifications caused by ReTIAR syndrome and DNA damage resulting from oxidative stress and action of heme products. In some cases, the “second hit” mutation in other genes (TP53, POLE, KRAS, PTEN, NOTCH1 or GATA) could initiate carcinogenesis. The niche of endometriosis SCs is modified by BMDSCs, disturbed function of galectins, chaperones, pro-inflammatory reaction and immune cells infiltrating endometriotic lesions. Epigenetic modulation of gene expression by DNA methylation and miRNA regulation could also push SCs toward malignant transformation. Extrinsic toxins and a bad diet could further modulate SCs and stimulate them to become CSCs. The hypothetical sequence of events influencing endometriosis stem cells, and leading from endometriosis to RAOC is presented in Figure 2.

## 15. Conclusions and Future Direction

Nowadays, treatment of endometriosis is a difficult task due to a high recurrence rate, technical obstacles of surgery and resistance to progestin therapy. Therefore, treatment of endometriosis needs complicated and sometimes disabling procedures. Drugs targeting the endometriosis epithelial stem cells or mesenchymal stem cells could improve the management and prevent disease recurrence. Such treatment would be especially welcome in the peritoneal, deep-infiltrating and recurrent ovarian endometriosis, where pain, adhesions and organ malfunction are mostly expressed. The anti-stem cell therapy has been slowly introduced into anti-cancer treatment and several pre- and clinical trials have been planned to estimate its efficacy. Due to the similarities between both the stem cells phenotype and function in endometriosis and cancer, there is a theoretical background to try anti-stem cell management in endometriosis. Some drugs directed against markers of stem cells (like CD117, CD133, EpCAM, ALDH1) have already been tested in ovarian cancer (imatinib mesylate, ALDH1 inhibitors, modified CAR-T lymphocytes) and could probably be tested in endometriosis where stem cells are characterized by the presence of the same markers. Simultaneously, reducing the risk of progression of endometriosis into EAOC could be obtained as a by-effect of stem cells-targeted therapy. The main problem that needs a solution is how to avoid an unwanted inhibition of stem cell-based regenerative function in eutopic endometrium. Moreover, the studies on a probable influence of environmental toxicity and improper diet on the progression of endometriosis could pave the way for epidemiological control of the disease. Improvement of the rules governing plant culture and animal breeding with the pressure to use more ecologic techniques could hopefully invert the present trend of an increase in female morbidity due to endometriosis.

## Figures and Tables

**Figure 1 cancers-15-00111-f001:**
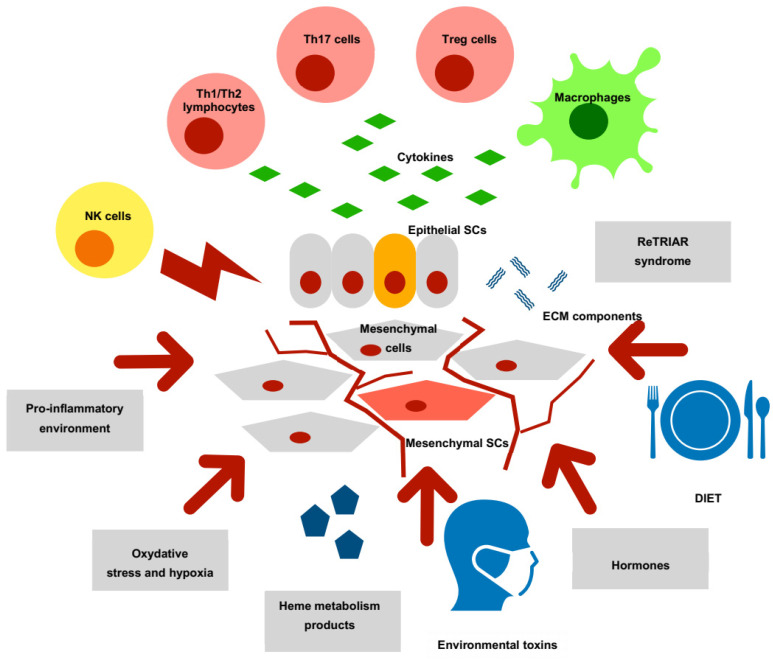
Niche for endometriosis epithelial stem cells Endometriosis epithelial stem cells (SCs) are regulated by several components of the endometriotic niche. Immune cells (natural killers (NK), Th1 and Th2 lymphocytes, Th17 lymphocytes, T regulatory cells (Tregs) and M1/M2 macrophages) are engaged in both elimination and supporting of endometrial lesions and their resultant action followed by secretion of cytokines and chemokines has a deep influence on survival of endometriosis epithelial SCs. Repeated injury and tissue repair (ReTRIAR) syndrome changes the ECM components and promotes fibrosis of implants. Fibrogenesis-mediated changes of gene expression, together with a pro-inflammatory environment could account for DNA damage in epithelial SCs. The mutational pressure of oxidative stress and heme metabolism products could depend on the fibrotic status of the endometriosis lesion. Oxidative stress, hypoxia and iron overload are highly mutagenic for epithelial SCs. Mesenchymal SCs are a relevant component of epithelial SCs niche. The participation of mesenchymal SCs in activin-A/ connective tissue growth factor (CTGF) pathway augments peritoneal inflammation and fibrosis in ectopic endometrial lesions. Disturbances in activin-A/CTGF pathway may be involved in carcinogenesis. Bone marrow-derived stem cells (BMDSCs) through cytokine secretion promote proliferation in endometriotic lesions. Estradiol is a hormonal mediator of mobilization of endothelial progenitor cells into endometrial implants, and in chemoattraction of BMDSCs into endometriosis implant’s stroma. Estradiol has also a stimulatory effect on inflammation and proliferation of implants. Finally, the components of diet and environmental toxins could modulate the function of endometriosis epithelial SCs.

**Figure 2 cancers-15-00111-f002:**
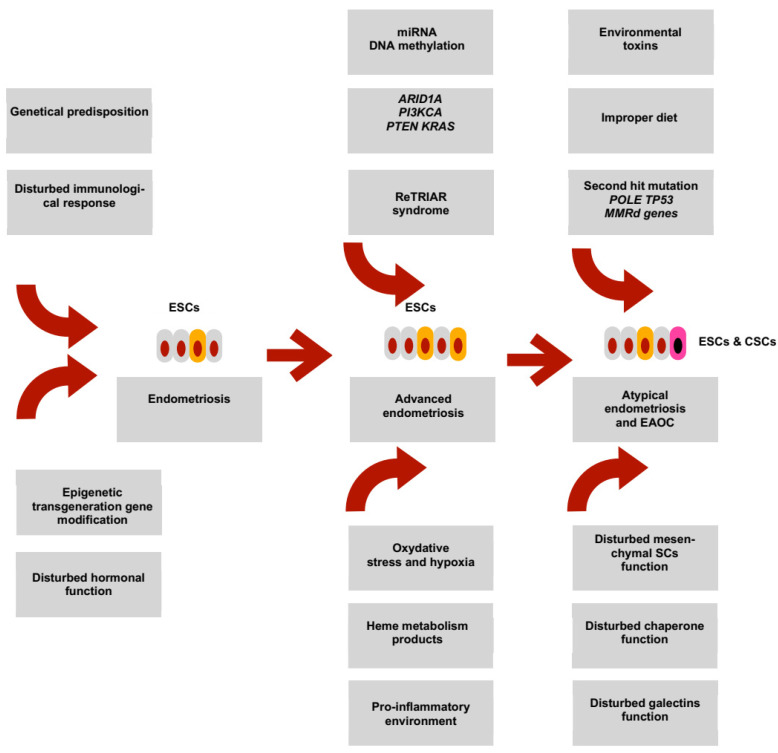
A hypothetical sequence of events leading from beginning of endometriosis towards its malignant transformation. In women presenting with a genetic predisposition or having epigenetic transgenerational gene modification endometriosis could emerge inside the peritoneal cavity, tubes and ovaries. Disturbed hormonal function (estrogenic stimulation) and ineffective immunological response enhance the chances for progressive growth of endometriotic lesions. Endometriosis epithelial stem cells (ESCs) originating either from endometrial stem cells or from bone marrow-derived stem cells are present inside endometriotic lesions. The prolonged existence of endometriotic foci leads to changes that could push ESCs to increased proliferative and functional activity. The mechanisms that contribute to these changes involve the occurrence of driver mutations (ARID1A, PI3KCA, PTEN, KRAS), ReTRIAR syndrome, epigenetic regulation by miRNA and DNA methylation. All these changes support the proliferative activity and migration potential of ESCs. Moreover, the DNA of the endometriotic ESCs could be damaged by a hostile environment composed of oxidative chronic stress, hypoxia, products of heme metabolism and a pro-inflammatory setting. These are known stimuli sustaining the function and survival of stem cells. Finally, in the group of elderly patients with non-treated endometriosis the progression into endometriosis-associated ovarian cancer (EAOC) takes place. The origin of EAOC is probably dependent on several trigger mechanisms influencing ESCs cells. One of the most probable events is so-called “second hit” mutation which destabilizes genetically ESCs with previous ARID1A or other mutations. Other triggers could be disturbed chaperone and galectins functions. One of the most relevant stimuli originates in our opinion from disturbed mesenchymal SCs function. These cells present inside the stroma of endometriotic lesions support proliferative activity and probably stemness of epithelial ESCs. Exposition to environmental toxins or improper diet components could be the final step in changing the ESCs into cancer stem cells (CSCs), and endometriosis into atypical endometriosis or EAOC.

**Table 1 cancers-15-00111-t001:** Stem cells in endometriosis and ovarian cancer and their role in both pathologies.

SCs Markers in Endometriosis		SCs Markers in Ovarian Cancer	
Eutopic endometrium of patients with endometriosis contains increases numbers of CD44^+^ cells. The BCL9/Wnt/CD44 axis is engaged in growth of endometriotic implants	[82] [83]	Higher expression of CD44^+^ cells in recurrent OC CD44v6^+^ OCSCs present in cancer with increased metastatic potential Patients with low-CD44v46^+^ tumors have better metastasis-free survival CD44^+^/CD24^+^/EpCAM^+^ cells show OCSCs properties with increased invasiveness and chemo-resistance	[126,127,144] [145]
CD117 is up-regulated in endometriosis implants	[84,85,86]	CD117^+^ OC cells correlated with resistance to chemotherapy and shorter recurrence -free interval CD44^+^/CD117^+^ cells are able to recapitulate tumors after transplantation into experimental animals	[128,129] [130]
CD133^+^Musashi-1^+^ cells were isolated from ovarian endometrioma	[146]	CD133+ correlated with cancer advancement, ascites, chemo-resistance Increased expression of Musashi-1 is correlated to unfavorable prognosis in OC patients More aggressive and advanced ovarian tumors have higher numbers of Musashi-1^+^ALDH1^+^ cells	[132] [137] [136]
Mesenchymal CD133^+^OCT-4^+^ALDH1^+^ stem cells were isolated from ovarian endometrioma	[147]	CD44^+^CD133^+^ALDH1A1^+^ OCSCs cells are present in chemo-resistant recurrent OC	[148]
ALDH1^+^ cells were present in endometrioma	[57]	ALDH1^+^ cells are OCSCs population possessing stemness properties, and being capable to restore the tumor ALDH-1^+^ cells were found in HGSOC and CCOC ovarian cancers, and were related to worse survival of patients	[149] [150]
Expression of SOX-2 is increased inside the stromal component of endometriotic lesion	[35]	Over-expression of SOX2 is related to stemness of cells and up-regulation of resistance to apoptosis.	[151,152]
Expression of OCT-4 and OCT-4 mRNA is increased inside the epithelial component of endometriotic lesion	[84,85,86]	Up-regulation of OCT4 in OCSCs was correlated to tumor progression and chemo-resistance	[153]
SUSD2^+^ mesenchymal SCs are more frequent in endometriotic lesions	[51]	SUSD2 expression in HGSOC was correlated with EMT, metastases and chemo-resistance	[143]
Mesenchymal SCs in endometriosis indicated increased expression of activin-A specific receptor and CTGF. Upon estrogen stimulation mesenchymal SCs showed increased expression of OCT-4, CD133 and ALDH1	[105] [152]	Up-regulated expression of activin-A and disturbed CTFG expression may be involved in carcinogenesis	[108,109]
Immunoreactivity of EpCAM epithelial cells is increased in ectopic compared to eutopic endometrium	[88]	EpCAM (+) OC cells have greater tumor-initiating potential compared to EpCAM (−) cells EpCAM expression is increased in chemo-resistant tumors	[133]

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
