# Peer review of "Endometriosis Stem Cells as a Possible Main Target for Carcinogenesis of Endometriosis-Associated Ovarian Cancer (EAOC)"

_cancers, 2022, doi:10.3390/cancers15010111_

Round 1

Reviewer 1 Report

Review by Wilczyński et al. is focused on stem cells in endometriosis and their potential role in carcinogenesis resulting into endometriosis-associated ovarian cancer development. It is well written comprehensive work dealing with topical topic. Thus, I recommend to accept this manuscript for publication in the journal after addressing several minor comments raised below.

1.       Difference among endometriosis, eutopic and ectopic endometrium should be somewhere, preferably in the introduction, clearly explained.

2.       Some descriptions in Figure 1 are missing.

3.       Can authors clarify or support the statement (line 446) “Galectins are gluten binding proteins…”?  

Author Response

Response to the P.T. Reviewer #1

We would like to thank you for your valuable remarks. According to them we have made some improvements to the manuscript:

1.       Difference among endometriosis, eutopic and ectopic endometrium should be somewhere, preferably in the introduction, clearly explained.

The difference between endometriosis, eutopic and ectopic endometrium has been explained, together with the meaning of the terms endometriosis implant, endometriosis focus and endometriosis lesion.  

2.       Some descriptions in Figure 1 are missing.

The missing description „Th1/Th2 lymphocytes” has been added

3.       Can authors clarify or support the statement (line 446) “Galectins are gluten binding proteins…”?

It should be of course „Galectins are glycan binding proteins…”. Unfortunately computer editor automatically changed the word into „gluten”. Now it has been corrected.

We once more appreciate your contribution to improvement of the manuscript. Thank you

Reviewer 2 Report

This review paper focuses on the role of endometriosis stem cells (ESCs) and proposes a possible sequence of events that ultimately leads to the development of endometriosis-associated ovarian cancers (EAOC). Although this paper provides a lot of interesting and useful information, it is not easy to read due to poor article layout, too few cohesive figures and tables, and too small font. It is recommended to edit the paper carefully. Here are some suggestions:

1. Some sections of this paper have a lot of content, and the relative relationship is very complicated, so it is difficult to imagine the correlations among different terms. It is suggested that the authors can add graphics to illustrate the correlations among these terms for the convenience of readers.

2. p. 1: The "Simple Summary" is recommended for removal as it is highly relevant to the Abstract.

3. Due to the use of many abbreviations in the text, for the convenience of readers, I recommended to list an abbreviation table at the end of this paper.

4. Line 249: I can’t find where the Figure 1 is.

5. p. 7: Table 1 should have its own title.

6. Line 615: The words „critical windows” should be “critical windows”.

7. Line 660: The words second hit” should be “second hit”.

8. Line 665: I can’t find where the Figure 2 is.

9. The conclusion section has insufficient relevance to the content of this paper, and I recommended to strengthen it.

10. There is a large gap between the format of the Reference section of this article and the requirements of the journal, and it needs to be carefully checked.

Author Response

Response to the P.T. Reviewer #2

We would like to thank you for your valuable remarks. According to them we have made some improvements to the manuscript:

Although this paper provides a lot of interesting and useful information, it is not easy to read due to poor article layout, too few cohesive figures and tables, and too small font. It is recommended to edit the paper carefully. 

Originally in the text there were 2 figures and 1 table. The font was originally Times New Roman 12. However, due to the editing process („Cancers” template, Zotero Editor template, original text in iMack Pages converted into Word) the text has been automatically formatted and some parts of it has been removed or changed. Sorry for this inconvenience. We had no idea that during the sending of the text to the „Cancers” it would change in that way.

1. Some sections of this paper have a lot of content, and the relative relationship is very complicated, so it is difficult to imagine the correlations among different terms. It is suggested that the authors can add graphics to illustrate the correlations among these terms for the convenience of readers.

The original 2 figures have been restored in the proper places of the manuscript in order to illustrate the correlations between the described mechanisms. Moreover, the text in the longer sections has been divided into sub-sections to introduce some clearance.

2. p. 1: The "Simple Summary" is recommended for removal as it is highly relevant to the Abstract.

The „Simple summary” has been re-written.

3. Due to the use of many abbreviations in the text, for the convenience of readers, I recommended to list an abbreviation table at the end of this paper.

The List of Abbreviations has been added at the end of the manuscript. The abbreviations were listed according to their appearance in the text and they were not repeatedly explained in the case when the explanation was primarily put into the text.

4. Line 249: I can’t find where the Figure 1 is.

The original Fig. 1 has been restored in the proper place of the manuscript

5. p. 7: Table 1 should have its own title.

The title has been added

6. Line 615: The words „critical windows” should be “critical windows”.

Has been changed, however, computer editor tries to restore the previous version

7. Line 660: The words „second hit” should be “second hit”.Has been changed

Has been changed, however, computer editor tries to restore the previous version

8. Line 665: I can’t find where the Figure 2 is.

The original Fig. 2 has been restored in the proper place of the manuscript

9. The conclusion section has insufficient relevance to the content of this paper, and I recommended to strengthen it.

The section has been re-written.

10. There is a large gap between the format of the Reference section of this article and the requirements of the journal, and it needs to be carefully checked.

The references have been created in the Zotero Editor according to the „Cancers” template, however, some of the journal names were indicated in the improper fashion. It has been checked and changed manually.

We would once more appreciate your contribution to improve the manuscript. Thank you.

Round 2

Reviewer 2 Report

Although this version of the manuscript has been improved greatly, some points still need to revise or modify, as follows:

1.      p. 10: Paragraphs in yellow background are not a correct sentence.

2.      p. 22, 23: The format of the second paragraph of Section 13 runs away.

3.      p. 25: The abbreviation table should follow the regular format, at least, should be arranged in alphabetical order.

4.      There are some text font and size errors in the text, please correct.

Author Response

Response to the P.T. Reviewer #2

We would like to thank you again for your valuable remarks. According to them we have made some improvements to the manuscript:

   p. 10: Paragraphs in yellow background are not a correct sentence.

It was a Tab.1 title I supposed - has been corrected

2.      p. 22, 23: The format of the second paragraph of Section 13 runs away.

Unfortunately I have no possibility to correct the spacing

3.      p. 25: The abbreviation table should follow the regular format, at least, should be arranged in alphabetical order.

The list has been re-organized in alphabetical order

4.      There are some text font and size errors in the text, please correct.

Have been corrected wherever noticed
